# Genome Identification and Expression Profiling of the *PIN-Formed* Gene Family in *Phoebe bournei* under Abiotic Stresses

**DOI:** 10.3390/ijms25031452

**Published:** 2024-01-25

**Authors:** Jingshu Li, Yanzi Zhang, Xinghao Tang, Wenhai Liao, Zhuoqun Li, Qiumian Zheng, Yanhui Wang, Shipin Chen, Ping Zheng, Shijiang Cao

**Affiliations:** 1College of Forestry, Fujian Agriculture and Forestry University, Fuzhou 350002, China; chuchu7613@163.com (J.L.); txh060404009@163.com (X.T.); lwh1623850793@163.com (W.L.); 13285946066@163.com (Z.L.); 19859795568@163.com (Q.Z.); chenshipin@fafu.edu.cn (S.C.); 2University Key Laboratory of Forest Stress Physiology, Ecology and Molecular Biology of Fujian Province, College of Forestry, Fujian Agriculture and Forestry University, Fuzhou 350002, China; 3FAFU-UCR Joint Center for Horticultural Plant Biology and Metabolomics, Haixia Institute of Science and Technology, Fujian Agriculture and Forestry University, Fuzhou 350002, China; zhangyanzi163@163.com; 4Fujian Academy of Forestry Sciences, Fuzhou 350012, China; 5College of Horticulture, Fujian Agriculture and Forestry University, Fuzhou 350002, China; w19819995339@163.com; 6Fujian Provincial Key Laboratory of Haixia Applied Plant Systems Biology, Center for Genomics and Biotechnology, College of Life Science, Fujian Agriculture and Forestry University, Fuzhou 350002, China; 7Pingtan Science and Technology Research Institute, College of Marine Sciences, Fujian Agriculture and Forestry University, Fuzhou 350002, China

**Keywords:** *Phoebe bournei*, *PIN* gene family, evolutionary analysis, expression profiling, abiotic stress

## Abstract

PIN-formed (PIN) proteins—specific transcription factors that are widely distributed in plants—play a pivotal role in regulating polar auxin transport, thus influencing plant growth, development, and abiotic stress responses. Although the identification and functional validation of *PIN* genes have been extensively explored in various plant species, their understanding in woody plants—particularly the endangered species *Phoebe bournei* (Hemsl.) Yang—remains limited. *P. bournei* is an economically significant tree species that is endemic to southern China. For this study, we employed bioinformatics approaches to screen and identify 13 members of the *PIN* gene family in *P. bournei*. Through a phylogenetic analysis, we classified these genes into five sub-families: A, B, C, D, and E. Furthermore, we conducted a comprehensive analysis of the physicochemical properties, three-dimensional structures, conserved motifs, and gene structures of the PbPIN proteins. Our results demonstrate that all *PbPIN* genes consist of exons and introns, albeit with variations in their number and length, highlighting the conservation and evolutionary changes in *PbPIN* genes. The results of our collinearity analysis indicate that the expansion of the *PbPIN* gene family primarily occurred through segmental duplication. Additionally, by predicting cis-acting elements in their promoters, we inferred the potential involvement of *PbPIN* genes in plant hormone and abiotic stress responses. To investigate their expression patterns, we conducted a comprehensive expression profiling of *PbPIN* genes in different tissues. Notably, we observed differential expression levels of *PbPIN*s across the various tissues. Moreover, we examined the expression profiles of five representative *PbPIN* genes under abiotic stress conditions, including heat, cold, salt, and drought stress. These experiments preliminarily verified their responsiveness and functional roles in mediating responses to abiotic stress. In summary, this study systematically analyzes the expression patterns of *PIN* genes and their response to abiotic stresses in *P. bournei* using whole-genome data. Our findings provide novel insights and valuable information for stress tolerance regulation in *P. bournei*. Moreover, the study offers significant contributions towards unraveling the functional characteristics of the *PIN* gene family.

## 1. Introduction

Plants experience various degrees of damage throughout their growth cycle in their natural habitats. Plant stress arises when the environmental conditions become unsuitable for optimal plant growth, which can be categorized into biotic and abiotic stress. Abiotic stresses include high salt, drought, low and high temperatures, flooding, mechanical damage, nutrient deficiency, oxidative stress, and so on. These factors impede plant growth and development, thereby limiting yield improvement, and can even result in plant mortality [1]. To overcome these adversities, plants have evolved defense mechanisms at the cellular, molecular, physiological, and biochemical levels to adapt to stress conditions [2,3,4]. *Phoebe bournei* (Hemsl.) Yang, belonging to the Lauraceae family and Phoebe genus, stands as a renowned arboreal species in the horticultural field. This species holds a prominent position among the flora within sub-tropical evergreen broad-leaved forests, boasting substantial economic worth and ecological relevance [5,6]. However, the growth trajectory of *P. bournei* has encountered impediments attributable to an array of anthropogenic interventions and abiotic stresses. Consequently, these challenges have induced a decline in its population size and have contributed to a dispersed distribution pattern in recent years [7]. The species is now regarded as a vulnerable species by the World Conservation Union and is a Class II protected plant in China. Therefore, it is imperative to enhance its tolerance to abiotic stresses.

Auxin, one of the earliest discovered plant hormones, plays a pivotal role in various aspects of plant development and growth processes [8], including cell differentiation, embryonic development, fruit ripening [9], oriented growth, and polar transport [10]. Auxin also exerts a crucial influence on the response of plants to stress, such as salt stress and low-temperature stress. Notably, auxin exhibits a distinct feature known as polar transport, which involves the movement of auxin between cells [11]. This polar transport is facilitated by the asymmetric distribution of PIN auxin output carriers on the cytoplasmic membrane [12,13,14]. The PIN protein, a membrane transport protein, exhibits a polar distribution within the cellular membrane and serves as the conduit for the intracellular to extracellular transport of auxin, making its polarity localization match that of auxin flow [15,16]. Recent research has substantiated that the polar transport of auxin in various organs is orchestrated through the interaction of *PIN* genes [17]. These proteins exhibit a polar distribution in the cell membrane and exert a significant influence on growth hormone transport. As shown in Figure 1A, the polar localization of PIN proteins is established through a process involving GNOM-mediated recycling and clathrin-mediated endocytosis (CME). This localization is maintained by clustering in the plasma membrane and through cell wall–plasma membrane connections. Apical–basal polarity is determined by reversible phosphorylation, which is regulated by PID/WAG kinases and PP2A phosphatases. The auxin transport activity of PIN proteins is mediated by D6PK. Additionally, PIN proteins undergo trafficking through the multi-vesicular body (MVB) for eventual degradation in the lytic vacuole [18,19]. Based on the specificity of their hydrophilic structure, PIN proteins can be classified into two categories. The first category is composed of short PINs, characterized by only one constant region 1, one variable region 1, and a short hydrophilic region. The second category encompasses long PINs, which exhibit an extended hydrophilic region containing three conserved C1-3 structural domains and two variable V1 and V2 structural domains [20]. Notably, a conserved NPXXY motif is presented between the hydrophobic and hydrophilic regions of the C-terminus [21], which plays an important role in lattice protein-dependent endocytosis and facilitates the interaction between receptor proteins and membrane proteins during growth hormone transport. The hydrophilic region of PIN proteins contains glycosylation and phosphorylation sites, which are closely associated with their functionality and proper localization.

To date, the *PIN* gene family has been extensively investigated in numerous species in terms of growth, development, hormones, and abiotic stresses [23,24,25]. In *Arabidopsis*, for instance, eight members of the *PIN* gene family have been identified, the physicochemical properties and functions of which have been thoroughly examined [26,27]. Among them, *AtPIN1-4* and *AtPIN7* are expressed on the plasma membrane and contribute to embryogenesis, while *AtPIN2* plays a crucial role in the redistribution of auxin in roots, particularly in response to gravitational forces [28,29]. In *Oryza sativa*, a total of 12 members of the *OsPIN* gene family have been characterized. The tissue-specific expression patterns of nine *PIN* genes have been investigated using RT-PCR and GUS reporter assays, revealing their up-regulation upon exogenous auxin stimulation [30]. However, the responses to different hormones varied among the different *PIN* genes [17,31]. In *Zea mays*, nine novel auxin efflux carriers belonging to the *PIN* gene family and two *PIN*-like genes have been identified. Further investigations unveiled overlapping expression domains of *ZmPIN* genes in the root apex, as well as during male and female inflorescence differentiation and kernel development [32]. Sixteen *ZaPIN* genes have been identified in the entire genome of *Zanthoxylum armatum*. In young leaves, most of the *ZaPIN*s exhibited up-regulation following stimulation with exogenous auxin and gibberellin [33]. In 2020, a total of 15 *PpPIN*s were identified in *Prunus persica*. Moreover, the expression patterns of these 15 *PpPIN* genes were analyzed in different cultivars, revealing differential expression among the various cultivars [34]. Additionally, the *MdPIN15* gene in *Malus domestica* and the *NtPIN4* gene in *Nicotiana tabacum* have been implicated in axillary bud formation [35,36].

*PIN* genes have also been implicated in the regulation of abiotic stresses; however, limited studies on their response to such stresses have been conducted. For instance, in *S. lycopersicum*, a total of 15 members of the *SlPIN* gene family have been identified, and the functional validation of genes exhibiting significant responses to adverse stress was carried out using gene silencing techniques [37]. In *S. bicolor*, *SbPIN3* and *SbPIN9* displayed high expression levels in flowers [38,39]. Moreover, 23 *GmPIN*s have been identified in *G. max*, with 15 of them found to participate in the response to drought stress. Experimental evidence has revealed that the majority of *GmPIN* genes exhibited down-regulation in r expression under drought stress [40,41,42]. In *Z. mays*, the expression levels of *ZmPIN1a* and *ZmPIN1b* were higher when compared to *ZmPIN1c* and *ZmPIN1d* under NAA and low phosphate treatment. Over-expression of *ZmPIN1a* and *ZmPIN1b* contributed to root development in transgenic lines, indicating the coordinated functions of *PIN1* genes during development and under abiotic stresses [43]. Furthermore, 44 members of the *TaPIN* gene family have been identified in *T. aestivum*, and RT-qPCR results demonstrated that several members of the *TaPIN* family can be concurrently induced in response to abiotic stresses [44]. In *L. chinense*, 11 *LcPIN* genes have been identified, with experimental data suggesting that *LcPIN5* and *LcPIN8* may play a crucial role in auxin transport in *L. chinense* stems and leaves under abiotic stresses [45]. Recent studies have indicated the presence of 16 *ZaPIN*s in *Z. armatum* and 12 *VvPIN*s in *V. vinifera*, and through stress experiments, it was observed that a majority of these genes responded to hormone stimulation or abiotic stresses in both species [33,46]. Accumulating evidence suggests that *PIN* genes, as a core part of signal transduction, play a key role in plant coordination and adaptation to multifarious abiotic stresses (Figure 1B).

In this study, we systematically analyzed 13 members of the *PIN* gene family in *P. bournei* using whole-genome data. In particular, we investigated the physicochemical properties of the encoded proteins and visualized their gene structures, chromosomal locations, and gene co-linearity. Additionally, an evolutionary relationship diagram was established between *PbPIN*s and *PIN* genes in other species. This study systematically examines the expression patterns of *PIN* genes in various tissues of *P. bournei* and their responses to abiotic stress, providing novel insights and information for future research on the selection and regulation of stress tolerance. Furthermore, it offers valuable insights and information for further understanding the functional characteristics of the *PIN* gene family.

## 2. Results

### 2.1. Identification and Physicochemical Properties of PbPIN Proteins

A total of 13 *PIN* genes were identified in the *P. bournei* genome. These genes were assigned names from *PbPIN1* to *PbPIN13*, according to their distribution on the seven chromosomes. The physical and chemical properties of the *PbPIN* genes were integrated and are presented in Table 1. The number of amino acids ranged from 180 (*PbPIN7*) to 632 (*PbPIN5*), and the relative molecular weight varied between 19,895.81 (*PbPIN7*) and 68,322.18 (*PbPIN5*). We found that *PbPIN6, PbPIN7,* and *PbPIN12* were hydrophobic proteins, while the remaining genes exhibited amphiphilic properties. Additionally, the projected sub-cellular localization indicated that the majority of *PbPIN* genes were located on the plasma membrane, with *PbPIN9* being the only gene located on the cytoskeleton. *PbPIN9*, whose isoelectric point is less than 7, is an acidic protein.

To further explore the protein structure of the *PbPIN* members, we employed SOPMA and SWISS-MODEL to predict their secondary and tertiary structures (Appendix A). Analysis of the secondary structure revealed that all *PbPINs* contained α-helices, extended chains, β-sheets, and random coils. Among the *PbPINs*, α-helices and random coils emerged as the primary secondary structure elements. The construction of three-dimensional models confirmed that the proportions of different structural components were consistent with the predicted secondary structures (Figure 2).

### 2.2. Phylogenetic Analysis of the PbPIN Gene Family

To investigate the evolutionary relationships between the *PbPINs* and *PINs* from other plant species, a phylogenetic tree was constructed with PbPIN, AtPIN, TaPIN, and OsPIN proteins using MEGA 7.0 (Figure 3). Based on the well-established classification in *A. thaliana*, the *PIN* gene numbers in *P. bournei* were classified into five major sub-families (A, B, C, D, and E) and seven classes (A, B1, B2, B3, C, D, and E). The B1 sub-family is composed of the highest number of *PbPIN* family members, including *PbPIN2*, *PbPIN6*, *PbPIN7*, and *PbPIN8*. Phylogenetic analysis revealed that *P. bournei* is closely related to *A. thaliana* and *T. aestivum*. It can be hypothesized that the *PIN* gene family in *P. bournei* has undergone relatively conserved evolution, retaining a larger set of intact *PIN* genes during its lengthy evolutionary process. The small number of *PbPIN* family members—each playing a crucial role in regulating growth hormone output—may be the main reason for the observed evolutionary classification.

### 2.3. Phylogenetic Analysis of the PbPIN Gene Family

We conducted a comprehensive annotation of the *P. bournei* genome and examined the chromosomal location of *PbPIN* genes in *P. bournei* (Figure 4). The distribution of *PIN* genes across 7 chromosomes of the *P. bournei* genome was observed to be uneven. As depicted in Figure 3, a total of 13 *PbPIN* genes were identified on 7 chromosomes: Chromosomes 1 and 3 harbored 3 *PbPIN* genes each, while chromosomes 8, 10, and 11 each contained a single *PbPIN* gene. Additionally, *PbPIN4* and *PbPIN5* were situated on chromosome 2, while *PbPIN9* and *PbPIN10* were located on chromosome 5.

The chromosomal localization data suggest that tandem replication and fragment replication events played significant roles in the evolution of *PbPIN* family genes. Such events have a notable impact on the amplification of genes related to abiotic and biotic stress responses. Collinearity analysis of the *PbPIN* gene family using TBtools revealed closely positioned pairs, such as *PbPIN1* and *PbPIN2*, as well as *PbPIN7* and *PbPIN8*, suggesting tandem duplication as a potential mechanism. Furthermore, three pairs of *PbPIN* genes (*PbPIN1* and *PbPIN12*, *PbPIN3* and *PbPIN4*, and *PbPIN5* and *PbPIN10*) exhibited collinearity, indicating mutual replication (Figure 5).

To gain further insights into the duplication events and potential evolutionary mechanisms of *PbPIN* genes, we generated comparative syntenic maps of *P. bournei* in comparison to five representative plant species, composed of three dicots (*A. thaliana*, *V. vinifera*, and *S. lycopersicum*) and two monocots (*O. sativa* and *A. comosus*); see Figure 6. Within these comparisons, *PbPIN* exhibited three pairs of homologous genes with *AtPINs* and *SlPINs*, four pairs with *VvPINs*, six pairs with *AcPINs*, and seven pairs with *OsPIN*s. These findings demonstrate that the genome collinearity between *P. bournei* and dicot plants surpassed that between *P. bournei* and monocot plants. Remarkably, *PbPIN5* displayed collinearity with genes from *A. thaliana*, *O. sativa*, *A. comosus*, and *V. vinifera*, indicating its widespread presence in monocot species and suggesting an ancient origin preceding species divergence. Furthermore, the *PbPIN12* gene exhibited a collinear gene in two distinct homologous groups within the *O. sativa* and *A. comosus* genomes, exclusively appearing in monocots. This observation suggests that gene duplication occurred subsequent to the divergence of monocots. Notably, *PbPIN5* and *PbPIN10* shared collinearity with the same gene loci as *O. sativa*, *A. comosus*, and *V. vinifera*, implying a common ancestral origin through gene duplication.

### 2.4. Protein Motif and Gene Structure Analysis of PbPIN Genes

Analysis of the conserved motifs within the 13 identified PbPIN protein family members unveiled the presence of 12 conserved motifs. Notably, members within the same sub-family exhibited consistent motif composition and sequential arrangement (Figure 7). Motif 8 was found in all genes except *PbPIN13*, highlighting its high conservation. The N-terminus of all genes, except for *PbPIN12* and *PbPIN7*, contained Motif 7, while Motif 3 was present at the C-terminus of all genes except *PbPIN13* and *PbPIN7*, which instead possessed Motif 5 and Motif 4, respectively. These observations indicate a relative conservation of *PbPINs*. *PbPIN13* and *PbPIN7* displayed only one and two conserved motifs, respectively, suggesting the potential loss or deletion of specific sequences during the evolutionary process. Furthermore, analysis of conserved domains revealed that the *PbPINs* harbored typical Mem_trans domains, except for *PbPIN13* and *PbPIN7*. Additionally, the *PbPINs* exhibited the Mem_trans superfamily, which closely resembles the Mem_trans domain in amino acid sequence and shares the function of membrane transport.

Gene structure analysis of *PbPIN* family members revealed varying numbers of exons (ranging from 4 to 7) and introns (ranging from 3 to 6), among which *PbPIN5*, *PbPIN7*, *PbPIN8*, and *PbPIN13* lacked untranslated regions (UTRs), while *PbPIN6* and *PbPIN10* lacked a 5’ UTR. The UTR in a gene sequence is known to significantly impact mRNA stability. Furthermore, differences in exon positions, exon numbers, and intron lengths were observed among different *PbPIN* gene family members, suggesting structural modifications or divergence within the family.

### 2.5. Multiple Sequence Alignment Analysis and Cis-Elements Analysis of PbPIN Genes

We conducted multiple sequence comparisons to confirm previous experimental validation and data analyses, which identified several functional elements and sites involved in regulating the polar transport and activity of PIN proteins (Figure 8). The structural domains of all PbPIN proteins, except for PbPIN1, PbPIN7, and PbPIN9, contained a cysteine residue (C). Phenylalanine residues (F) and the NPXXY element (found within Motif 6) were conserved in all PbPIN protein sequences. In addition, in our analysis of *P. bournei* protein sequences, we also identified the highly conserved TPRXS motif, suggesting the synergistic regulation of phosphorylation sites in *P. bournei*.

To gain further insights into the regulatory mechanisms of *PbPIN* genes and their response to plant hormones and stress, we conducted an analysis of the 2000 bp promoter sequence of *PbPIN* in *P. bournei* to identify potential cis-acting elements (Figure 9). This analysis revealed the presence of various elements associated with light response, hormone signaling, abiotic stress, and growth and development within the *PIN* gene family. Notably, hormone-responsive elements such as auxin- and methyl jasmonate (MeJA)-responsive elements were identified, along with elements responsive to salicylic acid, abscisic acid, and gibberellin. These results suggest that the *PbPIN* family genes may be more sensitive to stress and hormonal responses. The promoter region also contained elements related to abiotic stress factors, including anaerobic induction, defense and stress responses, low temperature responsiveness, wound signaling, drought response, and enhancer-like elements involved in anoxic-specific inducibility.

### 2.6. Expression Analysis of PbPIN Genes in Different Tissues

To gain deeper insight into the roles and regulatory mechanisms of *PbPINs* in the growth and development of *P. bournei*, we conducted an analysis of the expression patterns of the 13 *PbPIN* genes in various tissues, including the root bark, root xylem, stem bark, stem xylem, and leaf (Appendix A). The heat map analysis revealed distinct tissue-specific expression profiles of the *PbPIN* genes (Figure 10). Based on these patterns, the *PbPIN* genes were categorized into three branches. The first branch, consisting of *PbPIN9*, *PbPIN10*, and *PbPIN12*, exhibited predominantly elevated expression levels in leaves, indicating a potential association between these genes and leaf growth. Notably, all six genes in the third branch showed significantly high expression in stem bark, while four members of the second branch displayed pronounced expression in both root and root bark. In the stem bark, a majority of *PbPIN* members exhibited high expression levels, with some genes also showing expression in the leaves and root bark. In contrast, the xylem of the root and stem exhibited minimal or negligible expression for most *PbPIN* genes. These findings suggest that *PbPINs* may play regulatory roles in both root development and leaf growth processes.

### 2.7. The Expression Profile of PbPIN Genes under Abiotic Stress

To investigate the response of the *PbPIN* gene family to abiotic stresses such as drought, salt, and temperature stresses, we focused on five genes (*PbPIN1*, *PbPIN7*, *PbPIN8*, *PbPIN9*, and *PbPIN11*) that contained a higher number of elements associated with adversity in their cis-acting elements [22,47]. These five genes were considered to be representative of four different sub-families. Transcriptional analysis confirmed differential transient expression levels in response to different stresses.

The expression results, as depicted in Figure 11, revealed a distinct pattern for *PbPIN* genes under low-temperature stress at 10 °C, characterized by an initial up-regulation followed by a down-regulation. Specifically, *PbPIN1*, *PbPIN8*, and *PbPIN11* reached their peak expression levels at 6 h, 4 h, and 4 h, respectively, before gradually decreasing. At 40 °C, the expression levels of all genes also presented general up-regulation. Notably, when subjected to 10% PEG-induced drought stress, all five genes exhibited a cyclic rise-fall-rise-fall pattern, with significant peaks at 6 h and 12 h. In contrast, none of the five genes presented significant up-regulation under 10% NaCl-induced salt stress compared to the other three treatments. Instead, they displayed general down-regulation followed by up-regulation, ultimately resulting in a sharp decrease to nearly zero expression at 24 h.

## 3. Discussion

Since the initial discovery of the first PIN protein in *A. thaliana* [48,49], the identification of *PIN* gene family members has been carried out through whole-genome approaches in a range of diverse plant species. For this study, we employed systematic bioinformatic methods to conduct a comprehensive whole-genome identification and analysis of *PIN* genes in *P. bournei*. A total of 13 *PbPIN* gene family members were identified and extensively studied using the *P. bournei* genome database. Analysis of the physicochemical properties of these proteins revealed that, with the exception of *PbPIN9*, the theoretical isoelectric points (pI) of the remaining 12 *PIN* gene family members were all above seven. The analysis of physical and chemical properties showed that most of the *PbPIN* genes were alkaline amino acids, and each member contained the conserved domain Men_Trans (PF03547) in *P. bournei*, with consistent results in other plant species such as bamboo pepper [33] and soybean [40]. Moreover, *PbPIN7*, *PbPIN9*, and *PbPIN12* exhibited relatively shorter amino acid sequences, likely contributing to structural, functional, and property distinctions. Furthermore, the secondary structure prediction of these proteins demonstrated that all *PbPINs* are composed of α-helices, extended chains, β-sheets, and random coils. The α-helices and random coils represent the major secondary structural elements of the *PbPINs*, consistent with observations in bamboo pepper [33]. Tertiary structure prediction indicated a high similarity in 3D structure among proteins from different categories, except for *PbPIN7*, *PbPIN9*, and *PbPIN12*. This observation suggests a conserved protein structure level within the *PbPIN* gene family, while also indicating potential evolutionary changes and fragment deletions in *PbPIN7*, *PbPIN9*, and *PbPIN12*, corresponding to their relatively shorter amino acid sequences mentioned earlier, and it is speculated that this is also the basis for the location of *PbPIN9* in the cytoskeleton.

Systematic phylogenetic analysis revealed that, based on established evolutionary relationships in *A. thaliana*, the 13 PIN proteins from *P. bournei* can be classified into seven sub-families. Furthermore, compared to rice and wheat, the *PIN* gene family of *P. bournei* exhibited a closer homology to the *PIN* gene family members of *A. thaliana*, likely due to their shared characteristics as dicotyledonous plants. By analyzing the collinearity within the *PbPIN* gene family, we identified collinear *PIN* genes that arose through gene duplication events: *PbPIN1* and *PbPIN12*, *PbPIN3* and *PbPIN4*, and *PbPIN5* and *PbPIN10* (Figure 5). In rice, *OsPIN1b*/*OsPIN1d*, *OsPIN1a*/*OsPIN1c*, and *OsPIN3a*/*OsPIN3b* were similar in sequence, indicating that the *PIN* gene family was generated by duplication of chromosomal segments [18,19]. Motif analysis revealed that all *PbPIN* genes—except for *PbPIN13*—contained Motif 8, indicating its high conservation within the family. Moreover, different groups exhibited similar motifs, implying the functional significance of these conserved motifs. On the other hand, *PbPIN13* and *PbPIN7* possessed only one and two conserved motifs, respectively, suggesting that their sequences may have experienced losses or deletions during evolution. Such losses may be attributed to the selective loss of structural domains over the course of gene evolution or errors in annotation splicing [50,51]. Furthermore, analysis of the conserved domains revealed that the *PbPINs* harbor typical Mem_trans domains, except for *PbPIN13* and *PbPIN7*. Additionally, the *PbPINs* exhibited the Mem_trans superfamily, which closely resembles the Mem_trans domain in terms of amino acid sequence and shares the function of membrane transport. This superfamily includes growth hormone efflux carrier proteins and other transport proteins from diverse life domains. *PbPIN7* solely contained the Mem_trans superfamily domain, indicating a potential evolutionary change resulting in gene deletion. However, its function shared some similarity with other genes, in alignment with the findings of conserved motif analysis. In plants, introns play a pivotal role in regulating gene expression [52,53]. Further examination of intron–exon structures revealed that all genes contained both exons and introns, indicating a certain level of conservation in their structures. However, variations in the positions, numbers of exons, and lengths of introns were observed among different members of the *PbPIN* gene family, implying structural divergences or differentiations within the family. These findings further contribute to the understanding of their tissue-specific expression patterns [54,55].

PIN proteins play a crucial role as promoters of auxin efflux in the process of auxin polar transport, typically localizing to the plasma membrane and organelle membranes [56]. In our study, we predicted the plasma membrane localization of 12 *PbPINs*, suggesting their involvement in the transport of auxin from intracellular to extracellular regions [57]. Additionally, one *PbPIN* was predicted to localize to the vacuolar membrane, potentially contributing to cellular homeostasis by facilitating auxin flow between the cytoplasm and the vacuolar membrane [58]. The direction of auxin transport is determined by the phosphorylation status and polar localization of *PINs* [18,59]. Activation of auxin polar transport activity by PIN proteins is achieved through the binding of protein kinases to phosphorylation sites [60,61]. By aligning amino acid sequences, we identified highly conserved phosphorylation sites, TPRXS, and the NPXXY motif within the hydrophilic loop (HL) domain of all *PbPINs*. These elements contribute to the polar distribution of *PbPINs* in the plasma membrane and vacuolar membrane, thereby regulating auxin transport in plants. Notably, cysteine residues (C) were found in the structural domains of all PbPIN proteins, except for *PbPIN1*, *PbPIN7*, and *PbPIN9*. This motif was associated with the regulation of *PIN* activity and the control of polar *PIN* localization on the plasmalemma [62]. Furthermore, phenylalanine residues (F) were conserved in all PbPIN protein sequences, which have been shown in previous studies to interact with articulation proteins and play a role in the transport and polar localization of *PIN1* in *A. thaliana* [63]. Presumably, these residues also serve similar functions in *P. bournei*. The hydrophilic loop (HL) domains of PIN proteins contain motifs that play important roles in regulating the membrane abundance and polar localization of PIN proteins within the cell [64,65,66]. For example, the NPXXY element near the C-terminus is indispensable for *AtPIN1* localization [56], and this section also exhibited high conservation (found within Motif 6) in *P. bournei*. Previous studies have demonstrated that phosphorylation sites associated with these kinases are typically located within the highly conserved TPRXS motif. In our analysis of *P. bournei* protein sequences, we also identified the highly conserved TPRXS motif, suggesting the presence of synergistic regulation of phosphorylation sites in *P. bournei* [56]. Multiple sequence comparisons revealed the high conservation of these significant sites within the protein’s structural domains and in all motifs present in the *PIN* trans-membrane structural domain. This suggests that conserved loci in multiple motifs may perform similar functions across different species.

Previous studies have demonstrated the high sensitivity of *PIN* gene transcription to plant hormones and environmental conditions [20,67]. Promoter regions of *PIN* genes in various plant species have been found to contain numerous cis-acting elements associated with plant hormones (e.g., auxin, gibberellins, and abscisic acid) and stress responses (e.g., temperature, light, and drought) [32]. In our study, we identified multiple cis-acting elements related to plant hormones and stress responses within the promoter regions of *PbPIN* genes. Notably, the types and quantities of cis-acting elements differed among the various *PbPIN*s, suggesting that each gene exhibits a distinctive response to plant hormone treatments and environmental stimuli, including abiotic stresses. Various studies have demonstrated that several *PIN* genes in plants can be rapidly and concurrently induced in response to extreme stress conditions, indicating their involvement in stress regulation. Previous investigations have shown that soybean *PIN* genes are induced by various abiotic stresses and plant hormones, and their transcriptional responses to drought stress exhibit tissue-specific patterns depending on the severity of the stress [54]. Similarly, the up-regulation of *PIN* genes in response to plant hormone treatments and abiotic stress has been observed in *Z. mays* [30]. Furthermore, the auxin transport protein gene family in maize has been reported to respond to different abiotic stresses, with most members of the *ZmPIN* family exhibiting up-regulation in young leaves and down-regulation in roots under drought conditions [55].

Using a combination of phylogenetic analysis, transcriptomic data, and quantitative real-time PCR (RT-qPCR), we substantiated the swift induction and pivotal involvement of five representative *PbPIN* genes (*PbPIN1*, *PbPIN7*, *PbPIN8*, *PbPIN9*, and *PbPIN11*) in response to abiotic stresses, including low temperature, high temperature, drought, and salt stress. Our investigation revealed the up-regulation of these representative genes after exposure to specific stress conditions, including a low temperature of 10 °C, a high temperature of 40 °C, and drought stress induced by 10% PEG. Notably, *PbPIN1*, *PbPIN8*, and *PbPIN11* exhibited prompt induction in response to stress under low-temperature conditions, reaching peak expression levels within four to six hours. This observation suggests their potential involvement in the mechanisms underlying the tolerance of *P. bournei* to low temperatures. It has been reported that polar auxin transport is selectively inhibited by intracellular trafficking proteins—namely, auxin efflux carriers and influx carriers—under low temperature stress [68,69]. Consequently, it has been speculated that *PbPIN1*, *PbPIN8*, and *PbPIN11* may be responsible for impeding polar auxin transport under low-temperature conditions. Although no consistent patterns were discerned under salt stress, each gene exhibited distinct degrees of up- and/or down-regulation. Collectively, these findings provide preliminary evidence elucidating the rapid induction and vital roles played by the five *PbPIN* genes in response to abiotic stresses, including low temperature, high temperature, drought, and salt stress. Further investigations involving gene-specific over-expression or the analysis of *PIN* knockout plants are expected to hold promise in unraveling the precise functions of these genes.

## 4. Materials and Methods

### 4.1. Plant Material and Data Sources

The seedlings of *P. bournei* used in this research were generously provided by the Fujian Academy of Forestry. These seedlings were cultivated outdoors for a period of 10 months in red soil characterized by a pH of 5 and a soil organic matter content ranging from 2.57% to 6.07%. The growth area experienced an average annual temperature of 16–20 °C, accompanied by an annual precipitation ranging from 900 mm to 2100 mm and an approximate annual relative humidity of 77%.

The genome sequence data and annotation information for *P. bournei* were downloaded from the Sequence Archive of the China National GeneBank Database (CNSA), with accession number CNP0002030 [70]. Then, the protein sequences encoded by *PIN* genes were retrieved from the *A. thaliana* database (https://www.arabidopsis.org, accessed on 4 May 2022), previously established as a query sequence for the purpose of genetic screening and identification.

In terms of the expression profiles of *P. bournei*, we used the whole-genome data of *P. bournei* from the research group of Professor Zai-kang Tong. This data set provides transcriptome data analyses of *P. bournei* across five distinct tissues: stem bark, leaf, root bark, stem xylem, and root xylem. These data were obtained through RNA-seq analyses conducted on various tissues of *P. bournei*, sourced from the Bio Project database under the accession number PRJNA628065 [70].

### 4.2. Identification and the Physicochemical Properties of PbPIN Genes

A Hidden Markov Model of the conserved Mem_trans domain of the PIN (PF03547) was obtained from the Pfam database (http://pfam.xfam.org/, accessed on 31 January 2023), and the Simple HMM Search module in TBtools (version 1.108) was used to search the PIN protein sequences of the *P. bournei* genome. Then, 8 AtPIN protein sequences of *A. thaliana* were downloaded from the Plant Transcription Factor Database (PTFD; http://planttfdb.gao-lab.org/index.php, accessed on 31 November 2022). Using TBtools, we conducted a BLAST analysis of the entire *P. bournei* genome and extracted genes with e-values exceeding 10^−5^, resulting in a final set of 13 *PbPIN* genes. To verify that these candidates were *PbPINs*, the online website NCBI Conserved Domain Database (NCBI-CDD) (https://www.ncbi.nlm.nih.gov/cdd/, accessed on 31 May 2023) and SMART (http://smart.embl-heidelberg.de/, accessed on 31 May 2023) were further used to screen out the *PIN* sequences with Mem_trans domains. Following established conventions and based on extensive previous research, these genes were renamed accordingly. Physicochemical properties of the PbPIN proteins, such as isoelectric point, amino acid length, and molecular weight, were calculated using ExPASy (http://www.expasy.org/tools, accessed on 31 May 2023).

### 4.3. Motif Analysis and Gene Structure of PbPINs

The identified *PbPIN* sequences were then subjected to sequence alignment and motif analysis using the MEME tool. Motif identification was performed with settings including Zero or One Occurrence per Sequence (ZOOPS) and a maximum motif duplication limit of 10 and 3, respectively. Subsequently, we utilized the MEGA11 software for alignment and analysis, as well as generating informative logos, motifs, and distinctive structures.

### 4.4. Construction of the Evolutionary Tree

To establish connections between the *P. bournei* gene sequences and related *PIN* gene family sequences from *A. thaliana*, *O. sativa*, and *T. aestivum*, we employed MEGA7 with a bootstrap parameter set to 1000. An initial comparison of the evolutionary trees was conducted to ensure the desired correlation degree. The phylogenetic tree was further enhanced, and node distinctions were set using the evolve website.

### 4.5. Sequence Alignment and Three-Dimensional Structures of PbHsf Proteins

The conserved domains of PbPIN protein sequences were edited using the Jalview software. SOPMA (https://npsa-prabi.ibcp.fr/cgi-bin/npsa_automat.pl?page=npsa_sopma.html, accessed on 30 June 2023) was utilized for protein secondary structure prediction using the default parameters. Finally, the SWISS-MODEL database (https://swissmodel.expasy.org/, accessed on 30 June 2023) was used to predict the protein tertiary structures through the homology modeling method.

### 4.6. Promoter Cis-Element Analysis of PbPIN Genes

The putative promoter regions of *PbPIN* genes were analyzed for the presence of cis-acting elements using PlantCare (http://bioinformatics.psb.ugent.be/webtools/plantcare/html/, accessed on 31 July 2023), and the results were visualized using TBtools (https://www.omicstudio.cn, accessed on 30 July 2023).

### 4.7. Chromosomal Distribution and Synteny Analysis of the PbPIN Genes

The identified *PIN* genes were mapped to the chromosomes of *P. bournei* by comparing them to the available genome data for the species. Collinearity analysis was conducted using the TBtools software in order to elucidate any relationships between the homologous *PIN* genes in *P. bournei* and those in other selected species. To facilitate this comparative analysis, the whole-genome sequences and gene annotation files of seven species (*O. sativa*, *A. comosus*, *S. lycopersicum*, *A. thaliana*, and *V. vinifera*) were downloaded from the plant genome database. The resulting analysis atlas provided insights into the shared characteristics of the *PIN* gene family across these species.

### 4.8. The Expression Profiles of PbHsf Genes

Expression data for the *PIN* genes in *P. bournei* across various tissues were obtained from the Bio Project database (Appendix A). TBtools was employed to analyze this expression data and construct a gene expression heat map, offering a visual representation of the patterns and levels of gene expression.

### 4.9. Abiotic Stress Treatment

To ensure consistent growth potential, 1-year-old *P. bournei* seedlings with similar characteristics were selected. The materials were divided into a control group and a stress treatment group, consisting of 30 individuals and 3 individuals, respectively, for each treatment. Following the treatments, leaf samples were collected and immediately stored in liquid nitrogen at −80 °C for subsequent RNA extraction. The experimental treatments included simulated drought conditions, where control group seedlings were soaked in distilled water while the treatment group was exposed to a nutrient solution containing 10% PEG. Another group was subjected to salt treatment and soaked in a 10% NaCl nutrient solution. For the temperature treatments, the control group was maintained at room temperature, while the corresponding treatment groups were incubated at 40 °C or 10 °C. All samples were cultured in an artificial climate incubator with a temperature of 25 °C and a humidity of 75%. The treatment groups were sampled at 4 h, 6 h, 8 h, 12 h, and 24 h, while the control group was sampled at 0 h. To extract RNA, the collected leaves were ground, and a real-time fluorescence quantitative PCR (RT-qPCR) experiment was conducted to monitor the expression levels of target genes. A correlation clustering-labeled heat map was generated using the Spearman correlation algorithm in order to visualize the relationships between gene expression patterns.

### 4.10. RNA Extraction and qRT-PCR Analysis

Total RNA extraction was carried out using an RNA Extraction Kit (Omega Bio-TEK, Shanghai, China) for both the control and stress-treated samples. Following the manufacturer’s instructions, EasyScript One-step gDNA Removal and cDNA Synthesis SuperMix (Transgen, Beijing, China) were utilized to synthesize cDNA. Quantitative RT-PCR was subsequently performed using TransStart top green qPCR SuperMix (Transgen, Beijing, China). *PbEF1α* was employed as the internal reference gene [71], and the specific primers used in the experiment are provided in Appendix A. The mixture solution of the RT-qPCR reaction is composed of 1 μL of cDNA, 2 μL of specific primers, 10 μL of SYBR Premix Ex TaqTM II, and 7 μL of ddH_2_O. The RT-qPCR reaction process was as follows: pre-degeneration at 95 °C for 30 s; then 40 cycles of denaturation at 95 °C for 5 s; 60 °C for 30 s; 95 °C for 5 s; 60 °C for 60 s; and 50 °C for 30 s [72]. The relative expression of *PbPIN* genes was calculated using the 2^−ΔΔCt^ method, and one-way analysis of variance and Duncan multiple comparison tests were performed using the SPSS22.0 software, while GraphPad Prism8.0 was used for mapping [56]. To ensure robustness, all quantitative PCRs were conducted with three biological repeats and three technical replicates.

## 5. Conclusions

We conducted a comprehensive analysis of the *PIN* gene family in *P. bournei*, encompassing 13 members, using whole-genome data. Our analysis included an investigation of the physicochemical properties of the corresponding encoded proteins, visualization of gene structures, determination of chromosomal locations, and examination of gene collinearity. Additionally, we established an evolutionary relationship diagram comparing the *PbPIN*s with *PIN* genes from other species. By systematically analyzing the expression patterns of *PIN* genes in various tissues of *P. bournei* and their responses to abiotic stress, our findings provide novel insights and valuable information for investigations on selection and stress tolerance regulation. Moreover, our study provides significant contributions towards unraveling the functional characteristics of the *PIN* gene family.

## Figures and Tables

**Figure 1 ijms-25-01452-f001:**
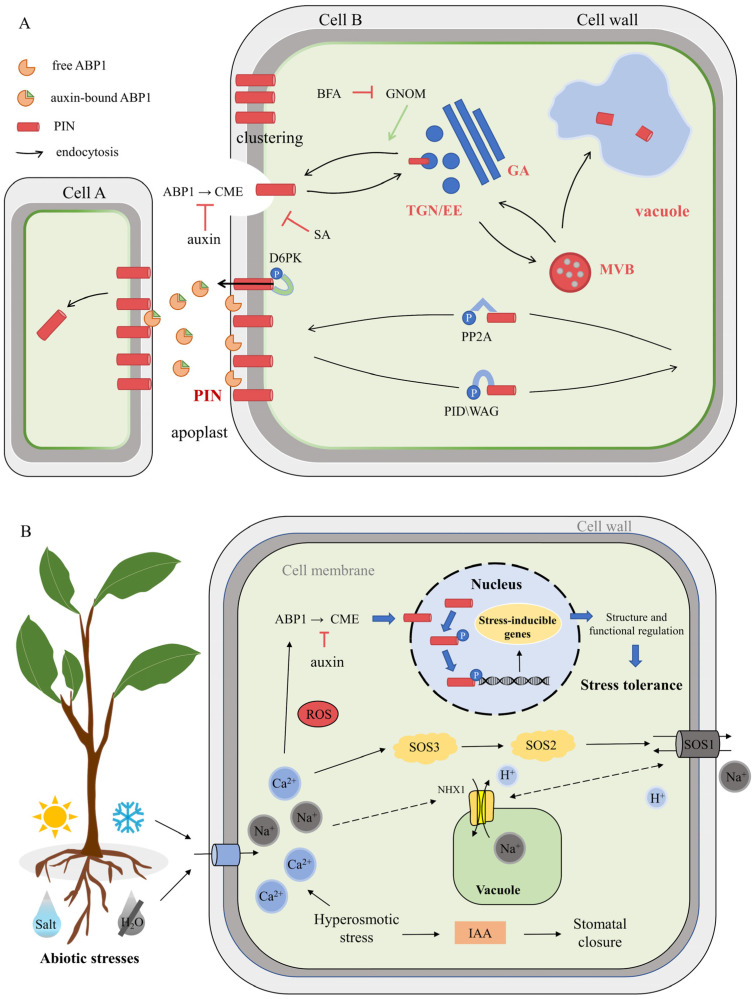
Schematic diagram of plant cell responses to stresses. (**A**) Sub-cellular trafficking and polarity maintenance of PIN proteins ( Adamowski et al. [18]; modified from Yin et al. [22]). GNO, guanylic acid exchange factor for ADP ribosylation factor; GA, Golgi apparatus; TGN/EE, trans-Golgi network/early endosome. (**B**) Working model of PIN transcription factor under abiotic stresses (heat, cold, salt, and drought). Solid arrows denote established positive effects, while dashed arrows indicate mechanisms that remain poorly understood. In the presence of abiotic stresses, cellular events such as elevated Ca^2+^ concentration, accumulation of reactive oxygen species (ROS), and protein degradation are initiated, ultimately transmitting the stress signal to the nucleus. Subsequently, PIN selectively binds to PLT/PID within the promoter region, activating the expression of stress-inducible genes. This concerted action enhances stress tolerance in plants.

**Figure 2 ijms-25-01452-f002:**
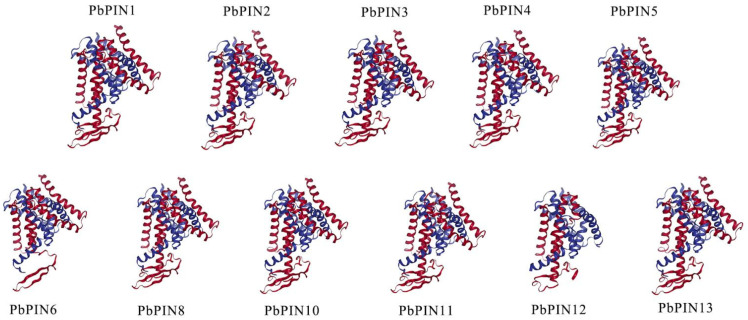
The predicted 3D structures of PbPINs. SOPMA was utilized for protein secondary structure prediction, using the default parameters. Finally, the SWISS-MODEL database was used to predict the protein tertiary structures through the homology modeling method.

**Figure 3 ijms-25-01452-f003:**
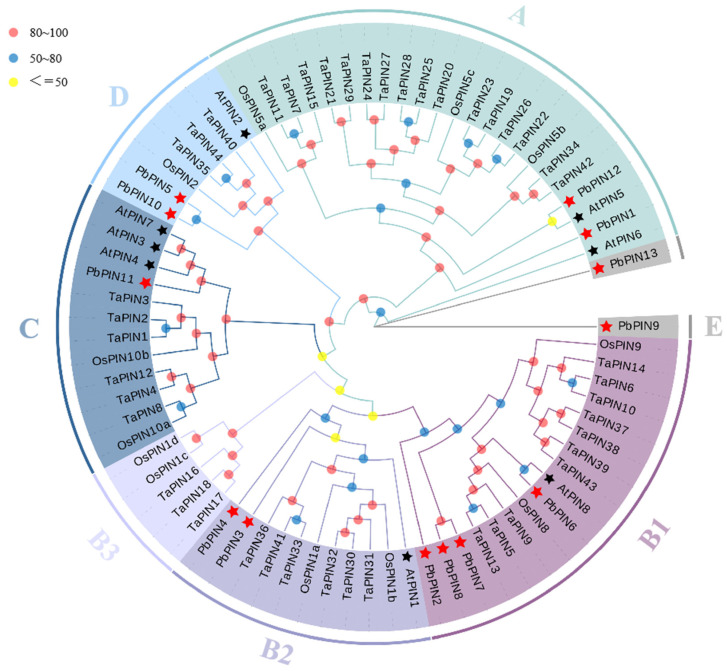
Phylogenetic analysis of the PIN protein sequences from *P. bournei* (Pb), *A. thaliana* (At), *T. aestivum* (Ta), and *O. sativa* (Os). The number on the branch denotes the reliability of the node based on 1000 iterations of Bootstrap verification. Branches of different classes have altered colors, each denoting a different sub-family, the red stars and black stars representing *PbPINs* and *AtPINs*, respectively.

**Figure 4 ijms-25-01452-f004:**
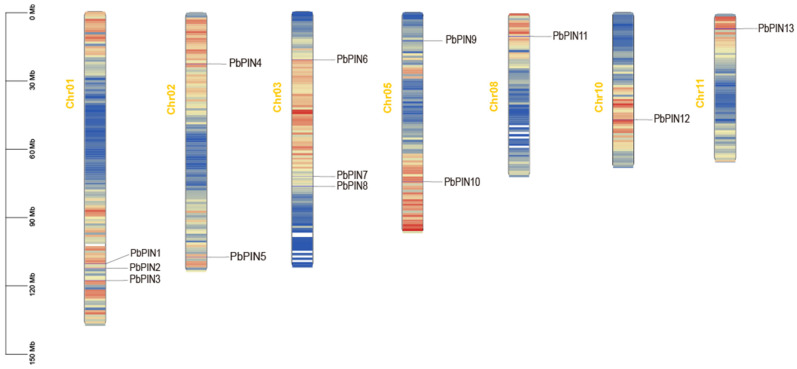
Chromosomal location of the identified *PbPIN* genes in *P. bournei*. The chromosomal location of the 12 mapped *PbPIN* genes is depicted from top to bottom. The scale bar is in Mb. Chromosome numbers are indicated on the left side of the corresponding chromosomes. chr: chromosome.

**Figure 5 ijms-25-01452-f005:**
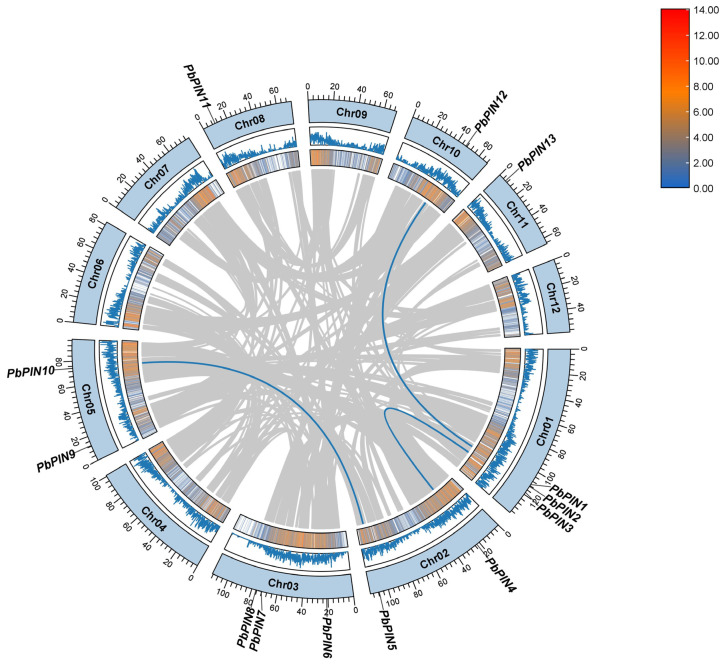
Chromosomal distribution and inter-chromosomal relationship of *PbPIN* genes. The two rings in the middle represent the gene density per chromosome, the gray line represents the collinear block in the genome, and the blue line represents the repeated *PbPIN* gene pair. TBtools was used for data processing.

**Figure 6 ijms-25-01452-f006:**
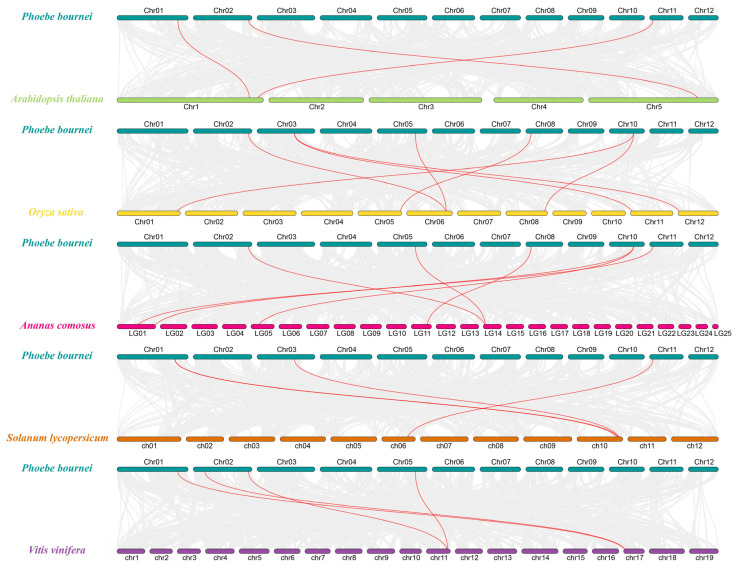
Homology analysis between the *P. bournei* genome and five plant genomes (*A. thaliana*, *O. sativa*, *A. comosus*, *S. lycopersicum,* and *V. vinifera*). The gray lines symbolize the aligned blocks between paired genomes, and the red lines indicate collinear *PbPIN* gene pairs.

**Figure 7 ijms-25-01452-f007:**
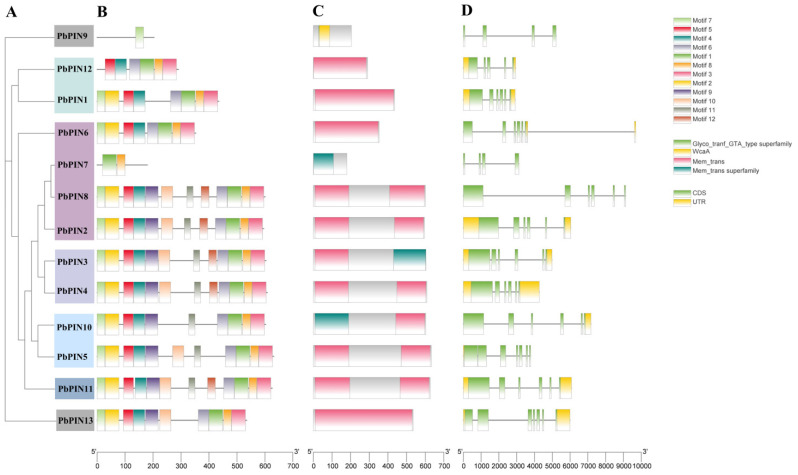
Structure of *PbPIN* gene members. (**A**) Phylogenetic relationships among *PbPIN* members. (**B**) Distribution of conserved motifs in the PbPIN proteins. A total of 12 motifs were identified. The scale at the bottom shows the length of the protein, and the sequence identity of each conserved motif is marked on the right. (**C**) Predicted conserved structural domains of PbPIN proteins. (**D**) Exon–intron structure of the *PbPIN* genes. Green boxes indicate exons (CDS), black lines indicate introns, and yellow boxes indicate 5′ and 3′ untranslated regions.

**Figure 8 ijms-25-01452-f008:**
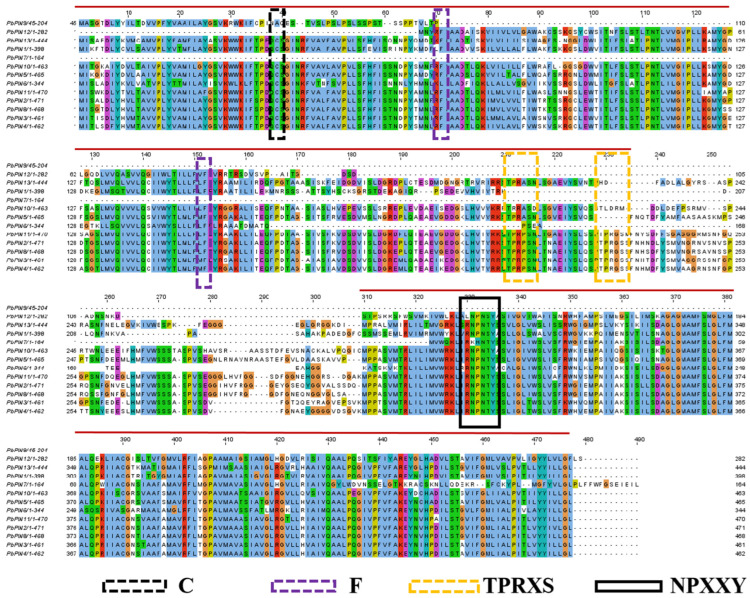
Multiple sequence alignments in the PbPIN protein sequence using the Jalview software. The red line indicates a conserved trans-membrane domain. Different amino acids are labeled with different color, and the possible functional sites or elements are encircled by a box. Letters with different colors represent different functions. C, cysteine; F, phenylalanine; T-S, phosphorylation region. Modified from Hu et al. [21].

**Figure 9 ijms-25-01452-f009:**
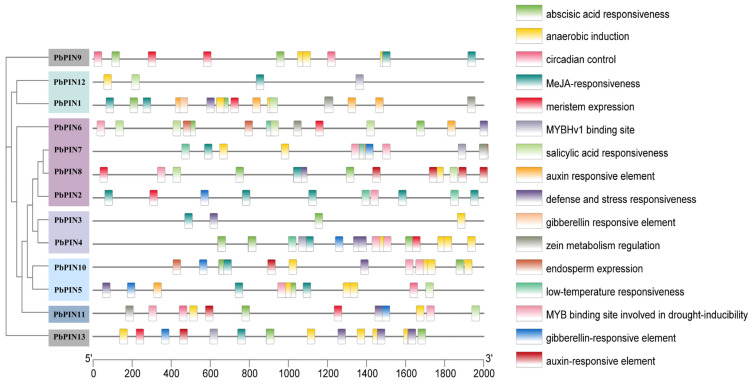
Predicted cis-acting elements in the promoter regions of *PbPIN* genes. On the left is the ML phylogenetic tree(bootstrap replications: 1000) and one on the right is the promoter position (0–2000 bp). The cis-acting regulatory elements in the promoter were categorized into 15 types with different colors. The lower axis denotes the gene length.

**Figure 10 ijms-25-01452-f010:**
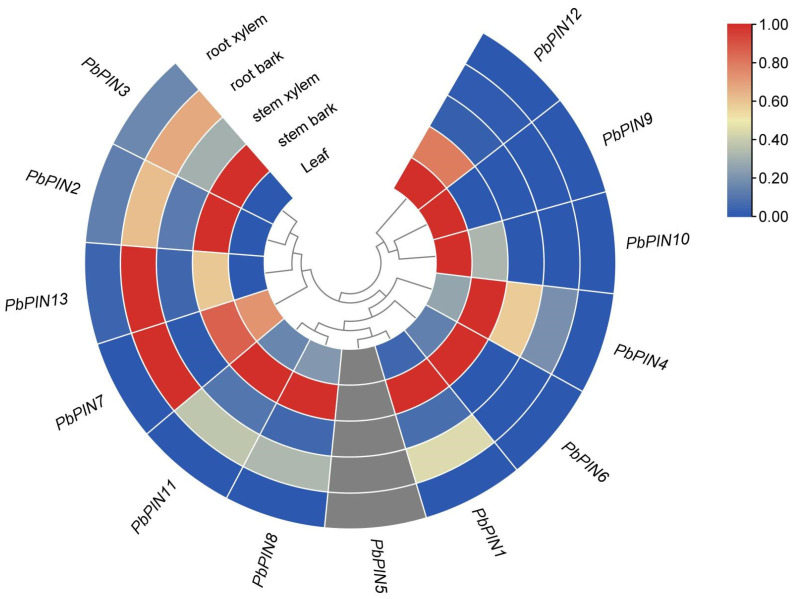
Tissue-specific gene expression patterns of 13 *PbPIN* genes. The expression patterns of genes in root bark, root xylem, stem bark, stem xylem, and leaf. Red and blue colors indicate high and low transcript abundance, respectively.

**Figure 11 ijms-25-01452-f011:**
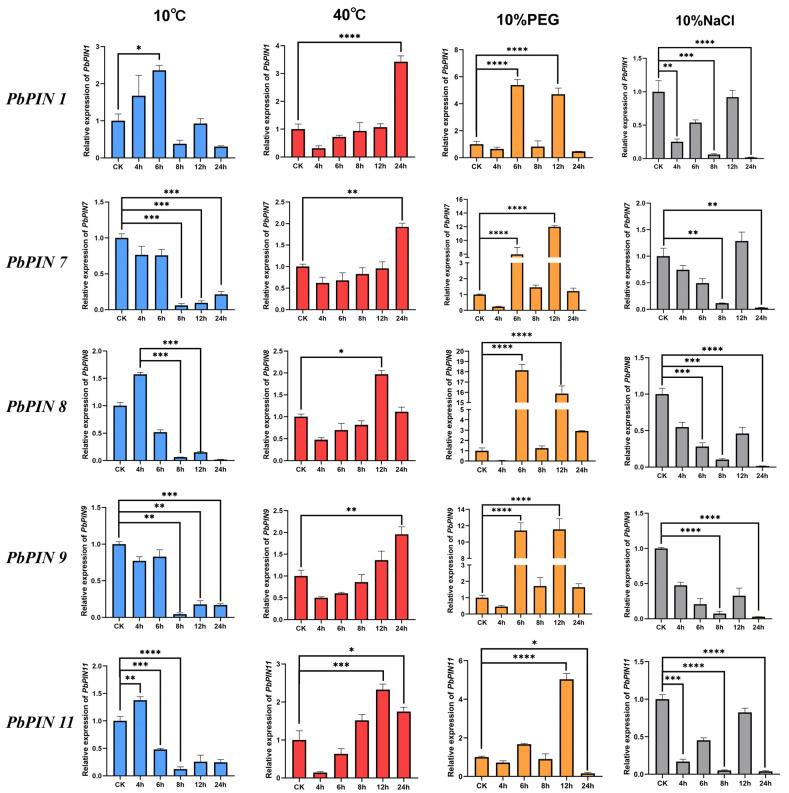
The differential expression of *PbPIN* genes in different tissues and the expression profiles of five representative *PbPIN* genes in response to different abiotic stresses, including cold, drought, salt, and heat stress. The relative expression levels of *PbPIN* genes in response to abiotic stresses assessed by RT-qPCR. Blue, low temperature; red, high temperature; orange, drought stress; gray, salt stress. The error bars indicate the standard deviations of the three independent RT-qPCR biological replicates. * represents a significant difference relative to the 0 h group (* *p* < 0.05, ** *p* < 0.01, *** *p* < 0.0005, **** *p* < 0.0001).

**Table 1 ijms-25-01452-t001:** Detailed information on 13 *PbPIN* genes in *P. bournei* and their encoded proteins.

Gene ID	Proposed Gene Name	Amino Acid Number	Molecular Weight	Theoretical Isoelectric Point	Grand Average of Hydropathicity (GRAVY)	Subcellular Localization
OF11384	*PbPIN1*	436	48,349.01	9.08	0.396	Plasma Membrane
OF11670	*PbPIN2*	596	64,931.38	9.13	0.221	Plasma Membrane
OF11843	*PbPIN3*	604	65,778.23	8.47	0.176	Plasma Membrane
OF03959	*PbPIN4*	609	65,773.99	8.81	0.187	Plasma Membrane
OF25220	*PbPIN5*	632	68,322.18	9.47	0.187	Plasma Membrane
OF12836	*PbPIN6*	353	38,684.47	9.50	0.727	Plasma Membrane
OF23475	*PbPIN7*	180	19,895.81	9.44	0.659	Plasma Membrane
OF28631	*PbPIN8*	601	65,380.68	8.91	0.192	Plasma Membrane
OF01149	*PbPIN9*	204	21,876.99	5.76	0.076	Cytoskeleton
OF26171	*PbPIN10*	603	65,672.23	7.56	0.300	Plasma Membrane
OF05766	*PbPIN11*	626	67,408.19	8.91	0.254	Plasma Membrane
OF06350	*PbPIN12*	291	31,377.13	9.17	0.745	Plasma Membrane
OF14189	*PbPIN13*	535	58,392.46	9.33	0.363	Plasma Membrane

## Data Availability

The genome sequence data and annotation information of *P. bournei* were downloaded from the Sequence Archive of the China National GeneBank Database (CNSA) with accession number CNP0002030.

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
