# Peer review of "Genome Identification and Expression Profiling of the PIN-Formed Gene Family in Phoebe bournei under Abiotic Stresses"

_ijms, 2024, doi:10.3390/ijms25031452_

Round 1

Reviewer 1 Report (Previous Reviewer 3)

Comments and Suggestions for Authors

The manuscript does not bring any real novelty to the PIN biology field. It is predominantly based on easily obtainable figures that are of a decent quality but of questionable value. MEGA, Circos, motif analyses, structures - all these outputs are readily available within a few hours, but the interpretation and the consecutive explanation of why these figures are important are missing. The illustrations would suit well a review article, but a research article must present some novel insights. 

1) What novelty and previously unknown regulations / conserved motifs were found by analyzing the PIN family in Phoebe bournei? The fact that this is not a model plant Arabidopsis does not justify the study unless some interesting novelty was found. 

2) There seems to be contradictory information in the figures. Figure 3 illustrates at least five Arabidopsis genes clustered with those of Phoebe bournei, but there are only three connections in genome analyses presented in Figure 6. 

3) The presented overview of promoter regions - is there a significant difference in cis-acting elements found in model plants? Is there some enrichment that is not found in A. thaliana / O. sativa? The information that PIN proteins are involved in hormonal signaling and growth/development is probably not that informative, as that is a well-known fact. 

4) Similar questions should be addressed with the reported expression profiles. The authors classified Phoebe bournei PINs (Figure 3). Do the corresponding orthologs in model species follow a similar response, or is that specific for Phoebe bournei

5) Figures 10 and 11 are missing statistical evaluation and contain incorrect statistics, respectively. For instance, it is impossible that PbPIN11 expression after 4h at 40°C is not statistically significant. 

6) There are mistakes in English. For instance, water deification stress could be an issue for religious debate, but it is definitely not part of plant molecular biology.  

Minor issues

The introduction is too lengthy for a research article.

In summary, the manuscript visual is shiny and looks decent, but the content itself seems too shallow for a Q1 journal.

Comments on the Quality of English Language

Needs some polishing

Author Response

Dear reviewer,

We feel great thanks for your professional review work on our manuscript. As you are concerned, there are several problems that need to be addressed. According to your nice suggestions, we have made corrections to our previous manuscripts, the detailed corrections are listed in attachment.

Thank you for consideration!

With best regards.

Reviewer 2 Report (New Reviewer)

Comments and Suggestions for Authors

The manuscript describes genome-wide identification of PIN proteins in P.bournei. The authors have performed adequate experiments and organized the manuscript well. However, few parts needs revision before considering the manuscript for publication.

1. In the genome-wide analysis results, the 3D structures of PIN7 and 9 is . missing in Fig2.

2. PIN9 is cytoskeleton bound according to the results. It will be good to discuss this in discussion section.

3. The expression of PINs under abiotic stress is well investigated however it will be more clear with a schematic representation based on the results something similar to Fig1 but with expression results with respect to abiotic stress related genes or pathways.

4. Minor grammatical / English revisions

Comments on the Quality of English Language

The manuscript is well written but minor grammatical errors can be revised

Author Response

Dear reviewer,

We feel great thanks for your professional review work on our manuscript. As you are concerned, there are several problems that need to be addressed. According to your nice suggestions, we have made corrections to our previous manuscripts, the detailed corrections are listed in attachment.

Thank you for consideration!

With best regards.

This manuscript is a resubmission of an earlier submission. The following is a list of the peer review reports and author responses from that submission.

Round 1

Reviewer 1 Report

Comments and Suggestions for Authors

As a reviewer, I find the manuscript's focus on the role of PIN-formed (PIN) proteins in polar auxin transport and their impact on plant growth, development, and abiotic stress response to be both relevant and engaging. The study's exploration of PIN genes, particularly in the context of woody plants like Phoebe bournei, fills an existing knowledge gap. The utilization of bioinformatics methods to uncover 13 members of the PIN gene family, their categorization, and subsequent analyses of their properties and structures is commendable. The findings on gene conservation, evolutionary changes, and collinearity patterns provide valuable insights into PIN gene expansion. The investigation into potential regulatory elements and the thorough expression profiling of PbPIN genes under diverse conditions add depth to the study. The preliminary evidence of PbPIN genes' involvement in stress responses is noteworthy. Overall, this manuscript is a good systematic analysis which enhances our understanding of PIN genes, their expression patterns, and their functional relevance in stress regulation in P. bournei. But i included following points for the improvement of the manuscript:

1.     Resolution of the figures like Figure 11 as the graphs were very difficult to understand as the number are not clearly visible.

2.     Also please add some information in the supplementary files as a header in the excel file sheet what that data is depicting.

3.     Furthermore, it is recommended to incorporate a note explaining the significance of asterisk (*) marks in the figures, clarifying their representation of statistical significance levels, such as p-values or other relevant indicators.

4.     If author can add some physiological assays data like chlorophyll content, gene expression analysis (pcr) with any abiotic stress marker genes then it can be a great add to manuscript.

Reviewer 2 Report

Comments and Suggestions for Authors

Comments:

In this manuscript entitled “Genome-wide Identification and Expression Profiling Of PIN-formed (PIN) Genes Family In Phoebe bournei Under Abiotic Stresses” authors have derived the PIN gene like sequences and studied their expression during the different abiotic stress treatments.

Title of this manuscript doesn’t not match, “genome wide identification…”. Thus study doesn’t show any novel approach or any new finding other than they pick few PIN gene like sequences and studied their expression. The findings in this study is still preliminary and not suitable to get published.

I recommend to reject this manuscript.

Comments on the Quality of English Language

Moderate editing of English language required

Reviewer 3 Report

Comments and Suggestions for Authors

The manuscript by Jingshu Li presents a descriptive study of PIN family genes in Phoebe bournei. The manuscript could be of interest, but its present form is not suitable for a Q1 journal.

Positives 

The study provides insight into the PIN family in a less-studied plant species, and further comparisons/analyses could identify as yet unknown mechanisms of PIN regulatory networks. 

Negatives

PIN proteins have been intensively studied for more than two decades, and the novelty in this work seems to be limited.

Major issues

1) Results presented in Figures 1-8 are based on relatively easily obtainable outputs of bioinformatics tools. It is a neat descriptive background, and most figures are of decent quality, but what is the point of the analysis? What is the hypothesis that is being addressed? Is there a significant deviation in the modeled structure/regulatory elements from that found in canonical PIN families in model species? And if so, is it possible to trace that difference to concrete physiological traits? How does this study improve our understanding of the PIN biology?

2) The authors indicate that the expression analyses were done in different tissues (Figure 10). The corresponding supplementary table is missing a description and it is not clear what is being presented in the figure/Supplementary Table S1. Statistical evaluation is missing. The legend of the heat map does not indicate transformation. Does that mean that 0 stands for not detected? How do the expression profiles match to the corresponding best-scoring orthologs from model plants?

3) Targeted analyses of five genes. The selection of these five genes for validating identified regulatory elements is flawed, as a negative control is missing. One of the genes with the lowest number of identified putative cis-acting elements should have been included in the study to provide evidence that the observed impact of stress is related to the number of identified elements. The authors should at least try to correlate observed regulations with the number of putative cis-acting elements. Furthermore, I have issues with Figure 11: It is of a lower resolution, and the labeling is illegible. Panel A - Images seem to be of a different scale, and it is not clear what is the "phenotypical" change. The changes in phenotype should be described in more detail, preferably substituted with a quantitative evaluation. Panel B - Statistical evaluation is insufficient. An ANOVA and a posthoc test (or similar test for time series comparison) should have been performed. The normalization factor is not clear. Some plots do not have 1.0 for the control sample. Fold change < 1.5 seems to be of a very low level for standard qPCR.

4) The manuscript should be modified to eliminate statements that are not supported by literature. Some examples - L49 - biotic stress is a common factor in nature; L49-L50 the list is not a complete list of all abiotic stressors - the language should be modified; L57 - what was the anticipated decline rate?; L74 - PIN are not general transporters of all growth hormones; 

5) The STRING interaction network. The description and legend for that figure are very poor. Furthermore, I don't believe that this analysis makes any sense. It is based on orthology with known genes and does not provide evidence that the given isoform interacts with the specific PIN in Phoebe bournei. How many isoforms are there for the identified putative interactions in the genome?

Minor issues

- Language/style requires polishing and editing, Latin names should be italicized.

- Some sections in results are part of the introduction, not results (e.g., L343-351).

- Description of all figures and tables must be improved to provide a clear indication of what is being presented. 

- Does the predicted localization in Table 1 match that of the corresponding orthologs in model plants? PbPIN9 is reportedly localized to the vacuole. What is the evidence for that?

- Legend for supplementary files is missing.

- Bioinformatics - the description of methods requires more detail to ascertain the reproducibility of the results by future researchers. 

In conclusion, this manuscript has potential, but I believe that it requires more work to be ready for publishing.

Comments on the Quality of English Language

language style and proofreading edits are needed

Round 2

Reviewer 3 Report

Comments and Suggestions for Authors

The revised manuscript has been improved, especially in grammar and style. However, the scientific part is still insufficient.

I will not list all the issues again, but these are the ones that I consider to be the most important:

1-(2) What is the hypothesis that is being addressed?

Response 1-(2): We would like to thank you pointing out this issue. It is widely recognized that PIN protein is a growth hormone transport carrier, but the mechanism of growth hormone transport is still unknown, and the analysis of the structure of PIN protein will provide us with powerful clues. Based on the bioinformatics approach, we mined the structure-function characteristics of PIN and elucidated them in an attempt to address the hypothesis of rapid induction and critical role of PbPIN gene in response to abiotic stresses such as low temperature and drought under different stress conditions. 

I am sorry to say that if that was the objective, it was not addressed at all. Further, I am surprised that the authors would consider PbPIN9 a real member of the PIN family. It does not show similarity with known PINs (Figure 2), its sequence is truncated and too short, and its structure does not seem to resemble a PIN protein. If this PIN protein was functional, it would be an interesting highlight of the study, but I don't believe that it fits the canonical PIN structure model (10.3389/fpls.2019.00985)

1-(3) Is there a significant deviation in the modeled structure/regulatory elements from that found in canonical PIN families in model species? And if so, is it possible to trace that difference to concrete physiological traits?

Not addressed in the revision - if there is no difference, the study does not make any new contribution.

2-(3) How do the expression profiles match to the corresponding best-scoring orthologs from model plants?

Response2-(3): In terms of the matching of the expression profiles of P. bournei and model plants, this study used the whole genome data of P. bournei. from the research group of Professor Zai-kang Tong, within this dataset, the transcriptome data analyses of P. bournei accross five distinct tissues including stem bark, leaf, root bark, stem xylem and root xylem. These data were obtained through RNA-seq analyses conducted on various tissues of P. bournei, sourced from the Bio Project database under the accession number PRJNA628065 [1]. Based on this, we conducted a comprehensive comparative study with the expression profiles of existing model plants, and conducted a special comparison and study of the key genes.

This response does not answer my question.

3-(3) Panel B - Statistical evaluation is insufficient. An ANOVA and a posthoc test (or similar test for time series comparison) should have been performed. The normalization factor is not clear. Some plots do not have 1.0 for the control sample. Fold change < 1.5 seems to be of a very low level for standard qPCR.

Response3-(3): Thank you for your advice. We have added the corresponding test methods in the paper. As well as changing the picture with higher resolution, it can be seen that the control sample is 1. (Please see line 622-625)

I don't see any difference in the revised figure, employed tests are not realistic. For instance, 24h  for PbPIN1 must be statistically significant unless the error bars were tempered with.

Description of all figures and tables must be improved to provide a clear indication of what is being presented.

Response : Thank you for your review and suggestions. We have improved the description of some of the charts.

Only a mild improvement, most legends are still insufficient. 

Comments on the Quality of English Language

Language still requires polishing. There are mistakes and incorrect phrasings, (e.g., Recent research has substantiated that the polar transport of auxin in various organs is orchestrated through the interaction of PIN genes - proteins are responsible for that, not genes), factual errors (e.g., deification stress), and stylistic (e.g., diagram of evolutionary relationship diagram).